# Effect of Coal Mining on Soil Microorganisms from *Stipa krylovii* Rhizosphere in Typical Grassland

**DOI:** 10.3390/ijerph20043689

**Published:** 2023-02-19

**Authors:** Linlin Xie, Yinli Bi, Yanxu Zhang, Nan Guo

**Affiliations:** 1State Key Laboratory of Coal Resources and Safe Mining, China University of Mining and Technology (Beijing), Beijing 100083, China; 2Institute of Ecological and Environmental Restoration in Mining Areas of West China, Xi’an University of Science and Technology, Xi’an 710054, China

**Keywords:** mining disturbance, arbuscular mycorrhizal fungi, soil fungi, microecological health, response strategy, edaphic variables

## Abstract

The environmental changes caused by coal mining activities caused disturbances to the plant, soil, and microbial health in the mining area. Arbuscular mycorrhizal fungi (AMF) play an important role in the ecological restoration of mining areas. However, it is less understood how soil fungal communities with multiple functional groups respond to coal mining, and the quantitative impact and risk of mining disturbance. Therefore, in this study, the effect of coal mining on soil microorganisms’ composition and diversity were analyzed near the edge of an opencast coal-mine dump in the Shengli mining area, Xilingol League, Inner Mongolia. The response strategy of soil fungi to coal mining and the stability of arbuscular mycorrhizal fungi (AMF) in the soil fungal community were determined. Our results showed that coal mining affected AMF and soil fungi in areas within 900 m from the coal mine. The abundance of endophytes increased with the distance between sampling sites and the mine dump, whereas the abundance of saprotroph decreased with the distance between sampling sites and the mine dump. Saprotroph was the dominant functional flora near the mining area. The nodes percentage of *Septoglomus* and *Claroideoglomus* and AMF phylogenetic diversity near the mining area were highest. AMF responded to the mining disturbance via the variety and evolution strategy of flora. Furthermore, AMF and soil fungal communities were significantly correlated with edaphic properties and parameters. Soil available phosphorus (AP) was the main influencer of soil AMF and fungal communities. These findings evaluated the risk range of coal mining on AMF and soil fungal communities and elucidated the microbial response strategy to mining disturbance.

## 1. Introduction

Coal mining has recently increased rapidly to support the economic development in China, but it results in the significant loss of biodiversity. Human activities such as coal mining and industrial emissions have been shown to be the main sources of soil pollution [1], which pose a notable threat to ecological security and human health. Different soil pollutions do not pose the same health risks [2]. Therefore, it is important to maintain the balance between economic development and healthy ecological environment. Studies have shown that the spatial distribution of soil and microorganisms and the interaction mode between microorganisms in mining areas are different from those in the non-mining areas [3,4]. In terrestrial ecosystems, soil microorganisms are important driving forces of biochemical reactions responsible for the decomposition of organic matter, nutrient cycling, and energy flow [2,5]. It has been shown that microorganisms in underground coal seams are potentially able to degrade pollutants produced during coal mining [6,7]. Biodegradation of coal by microorganisms could be a substitute for chemical combustion and a potential biocatalyst for coal transformation.

As a core component of microorganisms, soil fungi play an important role in soil improvement and plant litter decomposition [8,9]. Soil fungi are widely distributed and contribute to the transformation of soil nutrients [10]. The fungal communities developed stochastically during the ecological succession caused by environmental factors [11]. However, different fungi respond differently to abiotic factors, e.g., arbuscular mycorrhizal fungi (AMF) being more significantly linked to plant nutrient uptake under environmental stresses. Human activities, e.g., coal mining might affect biodiversity and ecosystem directly or indirectly. The sustainability of biodiversity relies on the stability of ecosystem itself [12].

AMF are important soil fungi living in the roots of most land plants symbiotically. It is well documented that AMF plays a positive role in increasing the absorption of plant nutrients, fixating carbon via photosynthesis, and enhancing the tolerance to biological [13] or abiotic (such as drought [14], heat [15], nutrients [16], and salt [17]) stresses. For instance, the extraradical mycelium of AMF is an important bridge for the transportation of nutrients outside the roots to the intraradical mycelium [18] and can promote the association of plants with bacteria and beneficial fungi [19]. Field studies by Bi et al. [20] and Xiao et al. [21] showed that inoculating plants with AMF increased soil enzyme activities and glomalin in the coal mining area. Meanwhile, AMF can alleviate the mechanical damages to the root system by simulating subsidence ground fissures due to coal mining [22]. These studies indicated that AMF play an important role in resistance to adverse conditions and vegetation optimization in the mining area. Colonization of AMF in plants can also be regulated by coal mining [23]. The mycorrhizal symbiosis formed by AMF and plants is a bridge connecting the aboveground and underground ecosystems. The ecological function of AMF is closely related to its diversity and composition [24], and diversity of AMF is essential to evaluating the effect of mine reclamation [25]. The function of AMF in ecological restoration has been widely recognized [21,26], whereas the role and stability of AMF in the whole soil fungal community were less understood, therefore investigations were necessary to better understand the microbial remediation and maintain the balance between ecological environment and mine production in the coal mining area.

Ecological stability has various definitions, e.g., resistance [27], resilience, or robustness [28,29] according to different research scales. Since environmental alterations can modify the structure of interaction networks [30,31], and network structure can in turn affect its stability (resistance and resilience to disturbance [32], biodiversity [33], and community stability [34]), the effect of coal mining on biological communities stability would be expected through microbial networks. Erkus et al. [35] showed that stability may increase the microbial communities with higher diversity. However, other studies showed the opposite perspectives to explain such changes. In the mining area, the soil fungal community is more stable than bacterial community because of the highly connected and complex ecological network (more positive co-occurrence relationships, and a greater average connectivity) despite the low species diversity [32]. Environmental disturbances can dramatically alter the network interactions among different microbial populations [36]. Although there is an association between microbial networks and stability, it is unclear how coal mining affects the connection between interaction patterns and fungal community stability. Elucidating network interactions of beneficial fungi (AMF) in soil fungal communities and their responses to environmental changes are fundamentally important for the understanding of microbial ecology, systems microbiology, and global change.

The objective of the present study was to determine the responses of microbial communities (AMF and soil fungi) to coal mining by collecting samples in the natural grassland sites at different distances from the waste dump of the mining area. We expect to address three key questions: how do AMF and soil fungal communities respond to coal mining, what factors driving changes of AMF and soil fungal communities, and what is the risk extent or range of soil AMF and fungi caused by coal mining disturbance?

## 2. Materials and Methods

### 2.1. Study Area and Sample Collection

The study was conducted at the periphery of the opencast coal-mine dump in the Shengli mining area, in the Xilingol League of Inner Mongolia, northern China. This site is situated at latitude 43°02′–44°52′ N, longitude 115°18′–117°06′ E with a typical climate of semi-arid grassland. The area has an average annual temperature of 0–3 °C, with considerable seasonal and diurnal temperature variation and mean annual precipitation of approximately 294.9 mm. The soil type of the experimental site is mainly meadow kastanozem. The area is dominated by the grasses of *Stipa krylovii*.

Samples were collected in the areas that were approximately 2 km in different directions from the dump, following a linear layout. Three sampling areas were selected in each of directions: route A, route B and route E. 100 m, 900 m, and 1900 m separated the dump from the three sampling sites on each route. Each distance has 3 replicate plots. Four subplots were established in each replicate plot with a total of 12 replicates at each distance. Five replicates of soil samples were randomly collected in a quadrat of 1 m × 1 m from the rhizosphere of native *S. krylovii* and then homogenized to obtain one final soil sample per subplot in July 2018. At a depth of 30 cm, samples of soil adjacent to the roots were obtained from each plot. A total of 36 soil samples and plants were collected and analyzed (3 distances × 12 duplicates). Fresh samples (10 g) were immediately processed for DNA extraction and high-throughput sequencing. Before processing for the analysis of soil physico-chemical properties, all the other samples were sieved (<2 mm), air-dried, and stored at 4 °C.

### 2.2. Analysis of Soil Properties

Soil pH and electrical conductivity (EC) in the soil environment may affect the growth, metabolism, and reproduction of microorganisms, while the growth and activity of microorganisms can also influence the soil pH, conductivity, and soil nutrients such as carbon, phosphorus, and potassium. Therefore, evaluating soil pH, conductivity, soil organic carbon (SOC), available phosphorus (AP), and available potassium (AK) is crucial for understanding microorganism activity. Soil pH and EC were measured using a glass electrode in a soil:water suspension [1:2.5 (*w*/*w*)] with a pH meter (PHS-3C; Shanghai Lida Instrument Factory, Shanghai, China) and in a [1:5 (*w*/*w*)] suspension with a conductivity meter (DDS-307W; Shanghai Lida Instrument Factory, Shanghai, China). The percentage of SOC used to estimate the soil organic matter (SOM) was determined by using a dichromate oxidation method in the presence of H_2_SO_4_ [37]. Soil AP and AK were assessed by inductively coupled plasma emission spectroscopy (ICP-OES, Optima 5300DV, USA). Soil phosphatase and urease activity were measured by the methods as described previously [38,39]. Glomalin, as a soil protein closely related to the activity of arbuscular mycorrhizal fungi, was quantified as a glomalin-related soil protein (GRSP) [40,41].

### 2.3. Soil DNA Extraction and High-Throughput Sequencing

Microbial DNA was extracted from 0.1 g of homogenized soil of each fresh sample using the MoBio PowerSoil DNA Isolation Kit (Mo Bio Laboratories, Carlsbad, CA, USA) according to manufacturer’s instructions. The internal transcribed spacer (ITS) between the large and small subunit rDNA was amplified by PCR utilizing the specific primers ITS1F (5’-CTTGGTCATTTAGAGGAAGTAA-3’) with a barcode and ITS2R (5’-GCTGCGTTCTTCATCGATGC-3’) [42]. The 18S rDNA was amplified (amplification conditions were similar to ITS rDNA) with the primers AMV4.5NF (5’- AAGCTCGTAGTTGAATTTCG-3’) and AMDGR (5’-CCCAACTATCCCTATTAATCAT-3’). PCR products were then mixed in equal density ratios and purified. DNA sequences were compiled and deposited in GenBank with accession numbers OQ136712-OQ136812.

### 2.4. Data Analyses

Raw sequences were analyzed by QIIME (v.1.7.0, http://qiime.org/ (accessed on 9 July 2020)). Sequences were grouped into operational taxonomic units (OTUs; ≥97% similarity) by UPARSE (v.7.0.1001, http://www.drive5.com/uparse/ (accessed on 9 July 2020)). The representative sequence for each OTU was screened for further annotation, and taxonomic information was obtained using Unite Database (https://unite.ut.ee/ (accessed on 9 July 2020)). The thorough bioinformatic analysis was reported previously [22]. Chao1, Shannon–Wiener index, Simpson index and phylogenetic diversity (PD) were used to assess species diversity, calculated, and displayed using R software (v.3.5.2, https://www.r-project.org/ (accessed on 10 June 2021) via the “microeco” package. Chord diagrams based on relative abundance of fungal communities were drawn using the “circlize” package. Principal Co-ordinates Analysis (PCoA) based on Bray–Curtis dissimilarity, Phylogenetic Analysis, and beta Net Relatedness Index (betaNRI) analysis were used to evaluate the differences in community composition among different distances performed using the “microECO” package. The SparCC-based fungal community structure network was used to evaluate the impact of sampling distance on the stability of the microbial community, analyzed by “SpiecEasi” package and visualized by Gephi (V.0.9.2, https://gephi.org/ (accessed on 10 June 2021)). Pearson analysis was performed to determine the correlations of each edaphic parameter with the distances, AMF, and fungal community composition. AMF and fungal diversity were analyzed by a Mantel test. Edaphic parameters were assessed by a one-way analysis of variance. Comparisons between pairs of means were analyzed using the Tukey’s honestly significant difference tests (*p* < 0.05) using SPSS (v.19.0, IBM, Armonk, NY, USA).

## 3. Results

### 3.1. Soil Properties Varied in Areas with Different Distances to the Coal Mining Area

Soil parameters were significantly different among areas with three distances (100 m, 900 m, and 1900 m) to the coal mining area (Table 1). Soil pH in the 1900 m area was significantly lower than that in the 100 m and 900 m areas (*p* < 0.05). The concentrations of EC, AK, SOM, and the activities of phosphatase and urease in the 1900 m area were significantly higher than those in the 100 m and 900 m areas (*p* < 0.05). Above results indicated that soil factors were influenced by mining activities in the vicinity of the mining area. The highest soil AP level was observed in the 900 m area (*p* < 0.05). The order of soil AP level in the three areas was 900 m area > 1900 m area > 100 m area. Interestingly, the level of easily extractable glomalin-related soil proteins (EE-GRSPs or EEG) was found to be not significantly different among the three areas, which might indicate that the activity of arbuscular mycorrhizal fungi was not affected by coal mining activities. 

### 3.2. AMF and Soil Fungal Composition and Function Varied among the Three Sampling Areas

A total of 109 AMF OTUs, 4 orders, 7 families, and 10 genus were identified with the predominant species being *Septoglomus viscosum* VTX00063 (24.55%), *Glomus* sp. VTX00167 (8.57%), *Glomus* sp. VTX00130 (6.93%), *Glomus* sp. VTX00100 (6.35%), *Glomus* sp. VTX00304 (6.23%), *Glomus* sp. VTX00214 (5.85%), *Glomus* sp. VTX00156 (5.84%), *Glomus* sp. VTX00222 (5.20%), *Glomus* sp. VTX00319 (4.74%), *Glomus* sp. VTX00409 (4.65%), *Claroideoglomus lamellosum* VTX00193 (4.13%), and *Glomus* sp. VTX00165 (2.53%) (Figure 1A). A total of 2544 soil fungal OTUs, 82 orders, 171 families, and 375 genera were identified with the predominant orders being Hypocreales (28.57%), Pleosporales (19.18%), Ascomycota (12.61%), Sordariales (12.20%), Agaricales (3.75%), Pezizales (1.74%), Mortierellales (1.54%), Capnodiales (1.51%), Eurotiales (1.35%), and Cantharellales (1.22%) (Figure 1B). 

Using a FUNGuild database, soil fungal OTUs were grouped into 10 functional groups with the major category being Plant Pathogen (15.05%), Undefined Saprotroph (12.22%), Endophyte (1.64%), Animal Pathogen-Endophyte-Lichen Parasite-Plant Pathogen-Soil Saprotroph-Wood Saprotroph (13.20%), Dung Saprotroph-Plant Saprotroph (6.63%), Animal Pathogen-Endophyte-Plant Pathogen-Wood Saprotroph (1.91%), Endophyte-Litter Saprotroph-Soil Saprotroph-Undefined Saprotroph (1.69%), Animal Pathogen-Dung Saprotroph-Endophyte-Epiphyte-Plant Saprotroph-Wood Saprotroph (1.50%), Endomycorrhizal-Plant Pathogen-Undefined Saprotroph (0.82%), Endophyte-Plant Pathogen-Undefined Saprotroph (0.62%), and the Unassigned category (44.72%) (Figure 1C). Plant Pathogen, Undefined Saprotroph, and Animal Pathogen-Endophyte-Lichen Parasite-Plant Pathogen-Soil Saprotroph-Wood Saprotroph showed the highest relative abundance in all three areas. The abundance of Endophyte and Animal Pathogen-Endophyte-Plant Pathogen-Wood Saprotroph increased with the distance to mining areas, whereas the relative abundance of Undefined Saprotroph and Animal Pathogen-Dung Saprotroph-Endophyte-Epiphyte-Plant Saprotroph-Wood Saprotroph decreased with the distance to coal mining area. The change of soil fungal community may be related to its functional adaptation after being disturbed by mining activities.

### 3.3. AMF and Soil Fungal α Diversities Varied among the Three Sampling Areas

Distance from a coal-mine dump significantly affected the α-diversity of the AMF community (Phylogenetic diversity) and soil fungal community (Chao1 index, Shannon Wiener index, Simpson index) (Figure 2). The AMF phylogenetic diversity at three areas ranged from 1.76 to 2.31 with the maximum being observed at 100 m area and minimum being observed at 900 m area, indicating that AMF at a 100 m area had more variability under environmental pressure. Other diversity indexes were not significantly different among the three areas, with the maximum observed at a 1900 m area (Figure 2A). The Shannon–Wiener index and the Simpson index with minimums being observed at the 100 m area and were significantly different from that in the 900 m and 1900 m areas. Shannon Wiener index was not significantly different between 900 m and 1900 m areas. The Chao1 index of soil fungi at the 100 m was significantly higher than that at 900 m (*p* < 0.05), indicating that certain fungal groups form a significant dominant community at the 100 m area. There was no significant difference in the phylogenetic diversity index among the three sampling areas in this study (Figure 2B).

### 3.4. Variation of AMF and Soil Fungal β Diversities among Different Sampling Areas

PCoA analysis visually showed the differences of AMF and soil fungal composition at the three different areas (Figure 3A,D), which was verified by the Bray–Curtis dissimilarity distance. The composition of the AMF community at the 100 m area was significantly reduced compared to that in the other two areas (*p* < 0.001), but there was no significant difference between the 900 m and 1900 m areas (Figure 3B). Based on the phylogenetic analysis, the beta net correlation index (betaNRI, Figure 3C) was calculated to reflect the response of different microbial communities to environmental pressure. The results showed that the betaNRI of AMF community at 100 m area was significantly higher than that at 900 m and 1900 m areas (*p* < 0.001), but it was not significantly different between 900 m and 1900 m areas (Figure 3C), indicating that the composition of AMF community was influenced by coal mining within 100 m, and the impact of coal mining was weak in areas with greater than 900 m distance. The Bray–Curtis dissimilarity distance and betaNRI of soil fungal community were similar with AMF (Figure 3E,F). Soil fungal community composition was affected at the 900 m and 100 m areas.

### 3.5. AMF and Soil Fungal Networks

Glomus, *Septoglomus,* and *Claroideoglomus* were the main groups of AMF network, accounting for 61.54–74.47%, 10.64–15.38%, and 4.26–7.69%, respectively (Figure 4). The nodes percentage of *Septoglomus* and *Claroideoglomus* was decreased with the distance to the mine having a maximum of 100 m. The maximum value of AMF network nodes appeared at 900 m, and the edges increased with the distance. The average path distances (GD) and modularity values of AMF networks (ranged: 3.326–4.45, 0.47–0.614, respectively) showed a decreasing trend with the increasing distance. The average degree (avgK) was opposite to the two indicators mentioned above. The average clustering coefficients (avgCC) variation ranged from 0.046 to 0.13, with a maximum value being observed at 900 m area (Table 2). Except for the unidentified groups, Sporormiaceae, Nectriaceae, and Pleosporaceae were the main groups of soil fungal network, accounting for 5.16–12.14%, 8.64–12.14%, and 5.00–5.56%, respectively. The number of nodes and edges of the soil fungal network both increased with the distance to the mine. Both avgK and avgCC had a maximum value at the 900 m area. GD ranged from 1.938 to 1.938 with a minimum value being observed at the 900 m area. The modularity values ranged from 0.170 to 0.214 and decreased with the distance to the mine.

Strikingly, the positive association of AMF with distance (100 m: 59.52%, 900 m: 66.2%, 1900 m: 52.78%) was much higher than that observed for soil fungi (100 m: 48.75%, 900 m: 50.12%, 1900 m: 50.23%). Particularly, in the near-mining area (100 m), the positive link of AMF is higher than the negative link, while the negative link of soil fungi was higher than the positive link, demonstrating that AMF had more positive co-occurrence relationships than soil fungi. Moreover, the modularity values of AMF network were larger, and were more connected and complex than those of soil fungi.

### 3.6. Correlation Analysis

Pearson’s correlation analysis and the Mantel test revealed significant associations between coal mining, soil variables, AMF, and soil fungal communities (Figure 5). Soil AP was negatively correlated with AMF phylogenetic diversity and soil fungal Chao1 index (*p* < 0.05), and positively correlated with soil fungal Shannon-Wiener index and Simpson index (*p* < 0.001). Soil pH was negatively correlated with soil fungal Shannon–Wiener index and Simpson index (*p* < 0.05). Soil SOM, AK, phosphatase, and urease activities were positively correlated with soil fungal α diversity (*p* < 0.05, Figure 5A). There were significant impacts of mining on soil parameters, especially soil pH, EC, AP, AK, phosphatase, urease, EEG (*p* < 0.01), and SOM (*p* < 0.05). Significant correlations were found between soil fungal composition and soil EC, AP, AK, phosphatase, urease (*p* < 0.01), and SOM (*p* < 0.05). Significant association existed between AMF composition and soil AK and urease activity (*p* < 0.05, Figure 5A).

Mining has significant impacts on soil factors, especially on soil pH, EC, AP, AK, phosphatase, urease, and easily extractable arbuscular mycorrhizal fungi spores (*p* < 0.01), as well as organic matter (*p* < 0.05). The structure of the soil fungal community is significantly correlated with EC, AP, AK, phosphatase, and urease (*p* < 0.01), as well as organic matter (*p* < 0.05). The structure of arbuscular mycorrhizal fungi (AMF) communities is significantly correlated with the activity of AK and urease in soil (*p* < 0.05). Therefore, mining disturbance may impact the structure of soil AMF and fungal communities through changes in AK and urease in soil. The Mantel test was used to quantify the relative effects of distance to the mine on soil parameters, composition of AMF and soil fungal communities (Figure 5B). The results showed that distance to the mine was significantly correlated with soil parameters (R = 0.583, *p* < 0.01) and composition of soil fungal community (R = 0.336, *p* < 0.01), but not with the composition of AMF community (R = 0.004, *p* > 0.05). These results indicated that the influence of coal mining on AMF was extremely weak, and, alternatively, AMF could be more stable to the disturbance of mining. There were significant interactions among soil parameters, soil AMF, and fungal communities. Soil AMF and fungal community composition might be driven by coal mining through soil factors.

## 4. Discussion

### 4.1. Response of Soil Fungi to Coal Mining

Environmental pressure can affect the stability of microbial community through the network of their interaction [43]. Soil quality and health can be reflected by microbial community, the diversity of which is closely related to community stability, e.g., community resistance and resilience. Therefore, elucidation of microbial diversity and community stability would promote our understanding on the mechanisms of microorganisms’ response to environmental pressure.

Hypocreales, Pleosporales, Ascomycota, and Sordariales are the predominant fungal orders in soil, with their relative abundance >10% (Figure 1). Hypocreales is an absolute dominant fungal group, and its lifestyle and host could change significantly during evolution. Interestingly, the predominant orders are also the common groups of dark septate endophytes (DSE) [44]. Root–fungal associations could be formed between host and DSE. Li et al. [45] showed that the symbiosis between DSE and plants depends on the specific plant symbionts. DSE can stimulate plant hormones and decompose carbohydrates, thereby providing monosaccharides for the host [46]. DSE can also promote a host’s absorption of water and nutrients and enhance the host’s tolerance to biological or abiotic stresses [47]. DSE can be pure cultured fungal groups and may play an important role in the ecological restoration of mining areas. 

Lower phylogenetic, Shannon-Wiener, and Simpson indices and higher Chao1 in the 100 m area indicate that predominant species might be formed near the mining area (Figure 2). The results indicate that in certain areas near the mining site, some fungi may form a significant dominant community, reducing the competitiveness of low-abundance species, thus reducing the diversity of fungal communities. According to the study of vegetation in the study area, the plant community tends to degenerate with the increase of mining disturbance [48]. Studies have shown that endophytes might change their nutritional mode and become active saprotrophs with the degradation or senescence of host [49,50]. Endophyte abundance increased with the distance, whereas the relative abundance of saprotroph decreased with the distance to the mining area (Figure 1). This might be due to the transformation of endophyte to saprotroph caused by the disturbance of coal mining, resulting in a significant increase of saprotroph as the dominant functional flora near the mining area.

### 4.2. Response of AM Fungi to Coal Mining

Soil microorganisms and their respiration can be inhibited or stimulated by soil pollution or interference. AMF can promote soil respiration and effectively indicate soil microbial activity under environmental stresses, which is closely associated with soil stability [51,52]. Elucidation of the stability of soil microorganisms at different distances from the coal mine would therefore promote our understanding of their resistance to coal mining. AMF phylogenetic diversity, soil fungal Shannon–Wiener index, Simpson index, and Chao1 index were significantly affected by coal mining (Figure 1). The change of soil fungal α diversity is stronger than that of AMF, indicating that AMF was more stable than soil fungi in response to coal mining. The networks of AMF and soil fungi demonstrated that AMF had more positive co-occurrence relationships and were more connected and complex than those of soil fungi (Figure 4). The Mantel test also suggested that the influence of coal mining on AMF was extremely weak, and AMF was more stable in response to mining disturbance (Figure 5). 

The influence of coal mining on AMF phylogenetic diversity is stronger than that on the Shannon–Wiener index, Simpson index, and Chao1 index (Figure 2). *Septoglomus* and *Claroideoglomus* increased significantly at 100 m area (Figure 4). AMF near mining area might have greater variability and evolutionary diversity. *Septoglomus* and *Claroideoglomus* could be considered indicators of coal mining disturbance. The increased tolerance mechanism of *Septoglomus* and *Claroideoglomus* induced by coal mining could well balance the competitiveness of dominant groups and improve microorganisms’ resilience and adaptability to environmental pressure.

### 4.3. Synergistic Interaction among Soil Properties, AMF, and Soil Fungi

In terrestrial ecosystems, many ecological and physiological processes (such as the turnover of organic matter, the effective regulation of mineral nutrients and the formation of mycorrhiza) are regulated by soil microorganisms [53]. Microorganisms can decompose plant litter and organic substances by secreting soil enzymes in the nutrient cycling process and promote plant growth by increasing soil available nutrients [54,55]. There is a reciprocal relationship between soil microbe communities and soil factors. Previous studies have shown that microbial activities are affected by changes in soil pH and moisture levels, and the increase of soil phosphorus has been proven to enhance soil microbial respiration [56,57]. The composition and diversity of AMF and soil fungal communities are both correlated with soil variables. Soil AK and urease activity play a critical role in AMF communities’ composition, while soil AP plays a significance role in determining AMF community diversity. The decomposition of SOM is a highly sensitive process that can lead to changes in community structure, particularly with the quantity and type of soil fungal communities. The allocation of organic carbon resources will shift with the change of fungal community composition [58,59]. Our study has revealed that SOM had a substantial impact on the dynamics of fungal communities, with SOM being positively correlated with fungal diversity. It is well known that the decomposition of organic substrates is a highly sensitive process to microorganisms, especially for the functional groups of soil fungi [60]. Here, the increase of SOM might be influenced by the functional groups of soil fungi. We hypothesize that the accumulation of SOM might be due to the increase in the abundance of Saprotroph, which enhanced the degradation of stubborn forms of carbon. The results of Looby et al. [60] support our findings, as they found that changes of fungal communities have a crucial impact on the decomposition of carbon compounds. Soil potassium is a key factor in determining plant resistance to stress factors such as cold, drought, and pests. During the decomposition of dead leaves, microbes produce a range of phenolic and chelate compounds, which promote the decomposition of potassium minerals in the soil and increase the levels of readily available potassium [61]. This study found that mining disturbance significantly reduced the levels of AK, which was significantly related to changes in the structure of both soil fungal and AMF communities, with a positive correlation with fungal community diversity. These results are consistent with the findings of Vuong et al. [62].

## 5. Conclusions

In this study, we comprehensively analyzed the microbial health under the disturbance of opencast coal mining. Our results demonstrated that soil AP was the key factor affecting soil AMF and fungal communities under mining disturbance. The soil fungal community responded to the mining disturbance through the strategy of endophyte-to-saprophyte transformation, whereas the AMF responded via the variety and evolution strategy of flora (*Septoglomus* and *Claroideoglomus*). Coal mining affects AMF and soil fungi in areas within 900 m from the coal mine. AMF were more stable in soil fungal communities. The stability of the AMF community could promote the resilience of the +soil microbial system and enhance its adaptability to environmental pressure. Hypocreales and Pleosporales, as the predominant soil fungal orders, are the common groups of DSE. Many similar unknown microbial resources need to be discovered. Specifically, the conservation and the development of beneficial fungi targeting specific ecological functions are important for biological reclamation and environmental health of the mining areas. Limited by the time constraints of sampling, the potential influence of extreme weather conditions on the results cannot be excluded. Therefore, conducting sampling with multiple years and seasons is a priority for future research. Additionally, potential changes in fungal function as results of mining disturbances in our study were observed, while it is difficult to elaborate in detail through amplicon sequencing methods. In the future, we will use metagenomics or metabolomics methods to provide a more detailed explanation of the mechanisms of fungal function changes.

## Figures and Tables

**Figure 1 ijerph-20-03689-f001:**
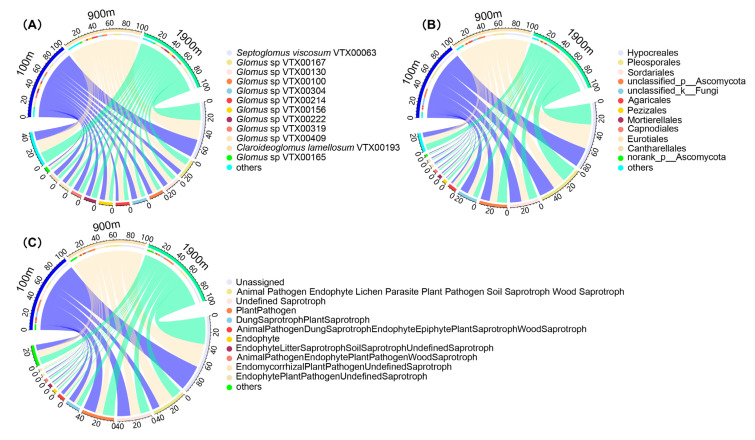
Proportion and function of AMF and soil fungal communities. (**A**): Soil AMF community composition, (**B**): Soil fungal community composition, (**C**): Soil fungal community function.

**Figure 2 ijerph-20-03689-f002:**
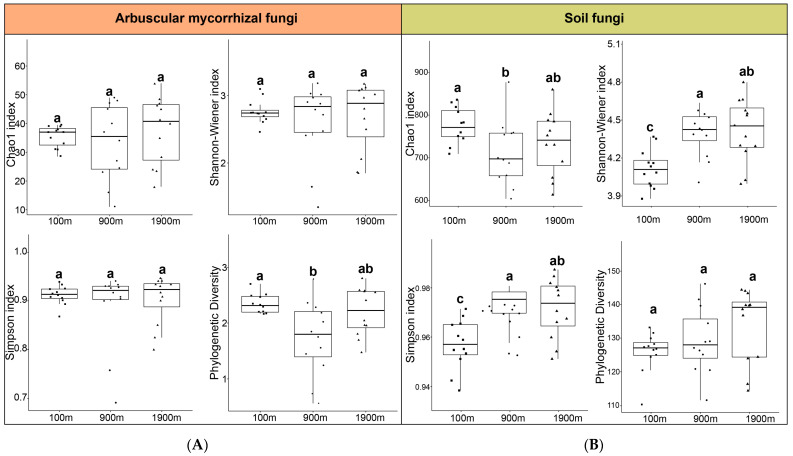
Effects of distance from coal-mine dump on the α-diversity of AMF (**A**) and soil fungal (**B**) communities. Rectangles, dots and triangles represent the values of diversity indices of replicate samples at different distances, respectively. Significantly different are bars followed by lowercase letters (*p* < 0.05).

**Figure 3 ijerph-20-03689-f003:**
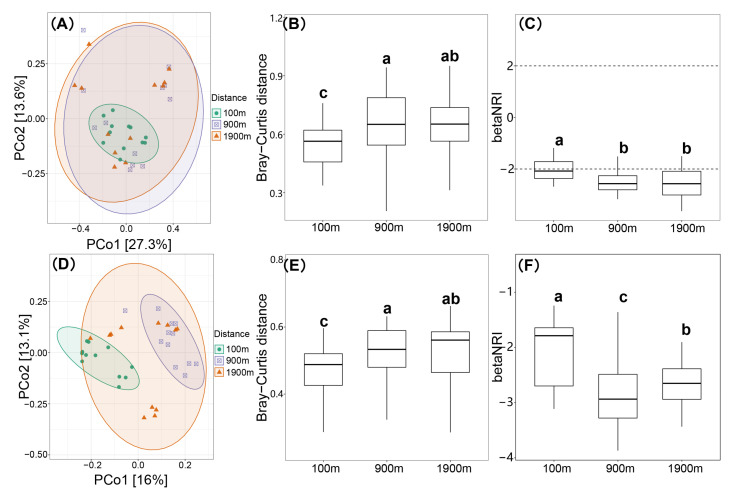
β diversity analysis of soil AMF and soil fungal communities. (**A**): Soil AMF Principal Co-ordinates Analysis (PCoA), (**B**): AMF Bray–Curtis distance, (**C**): AMF β-Net Relatedness Index (betaNRI), (**D**): Soil fungal PCoA, (**E**): Soil fungal Bray–Curtis distance, (**F**): Soil fungal betaNRI. Significantly different are bars followed by lowercase letters (*p* < 0.05).

**Figure 4 ijerph-20-03689-f004:**
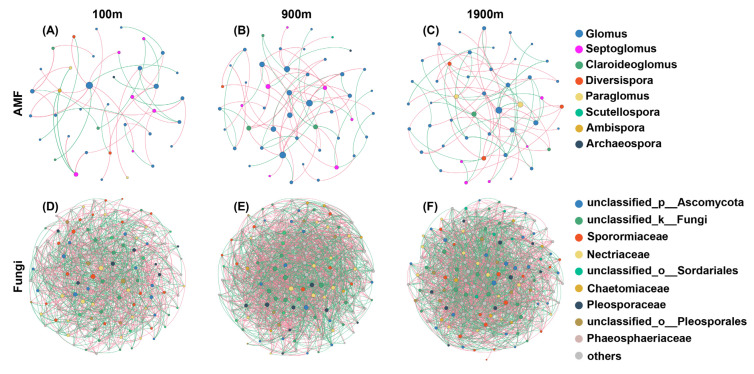
Networks of soil AMF and fungal communities. The nodes are colored according to the taxonomic information at genus (AMF) and family (soil fungi) level. Lines connecting nodes represent positive (red) or negative (green) co-occurrence relationships. (**A**): network of soil AMF at 100 m; (**B**): network of soil AMF at 900 m; (**C**): network of soil AMF at 1900 m; (**D**): network of soil fungi at 100 m; (**E**): network of soil fungi at 900 m; (**F**): network of soil fungi at 1900 m.

**Figure 5 ijerph-20-03689-f005:**
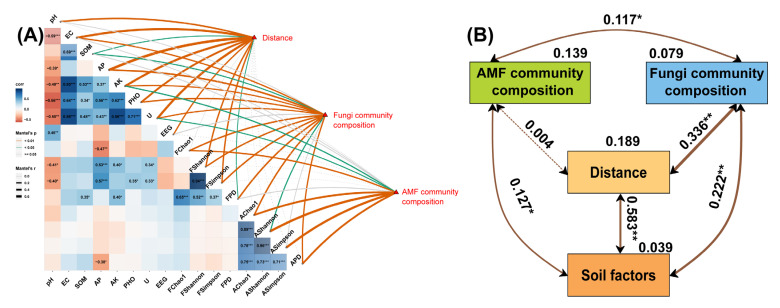
Relationships among distances, edaphic variables, AMF, soil fungal diversity indices, and composition of AMF and fungal community. Pairwise comparisons of edaphic variables and fungal indices with a color gradient denoting Pearson’s correlation coefficient in (**A**) (* *p* < 0.05, ** *p* < 0.01, *** *p* < 0.001). EC: electrical conductivity, SOM: soil organic matter, AP: available phosphorus, AK: available potassium, PHO: phosphatase, U: urease, EEG: easily extractable glomalin-related soil protein. FChao1: soil fungal Chao1, FShannon: soil fungal Shannon-wiener index, FSimpson: soil fungal Simpson index, FPD: soil fungal phylogenetic diversity, AChao1: soil fungal Chao1, AShannon: AMF Shannon-wiener index, ASimpson: AMF Simpson index, APD: AMF phylogenetic diversity. Solid lines and dashed lines indicate significant and non-significant pathways in (**B**), respectively. The width of the solid lines indicates the strength of the causal effect, and the numbers near the arrows indicate the value of R (* *p* < 0.05, ** *p* < 0.01).

**Table 1 ijerph-20-03689-t001:** Changes of soil parameters at areas with different distances to the mine.

Distances	100 m	900 m	1900 m	F	*p*
pH	7.62 ± 0.07 a	7.58 ± 0.03 a	7.52 ± 0.05 b	11.2	<0.001
EC (μs·cm^−1^)	251.33 ± 3.65 b	246.79 ± 2.33 c	353.58 ± 2.75 a	4991.5	<0.001
AK (mg·kg^−1^)	157.88 ± 4.13 b	160.27 ± 2.94 b	170.67 ± 1.97 a	56.29	<0.001
AP (mg·kg^−1^)	3.93 ± 0.10 c	4.9 ± 0.21 a	4.68 ± 0.11 b	139.99	<0.001
SOM (mg·g^−1^)	34.55 ± 0.97 b	33.22 ± 0.86 c	36.15 ± 1.40 a	21.3	<0.001
Phosphatase (µg·g^−1^·h^−1^)	191.16 ± 1.83 c	196.67 ± 4.7 b	203.09 ± 5.94 a	21.2	<0.001
Urease (μg·g^−1^·h^−1^)	52.35 ± 2.71 b	54.43 ± 2.75 b	62.75 ± 1.47 a	64.01	<0.001
EEG (μg·g^−1^)	50.46 ± 0.12 a	50.41 ± 0.03 a	50.43 ± 0.04 a	1.43	0.253

EC, electrical conductivity; AP, available phosphorus; AK, available potassium; SOM, soil organic matter; EEG, easily extractable glomalin-related soil protein. Different lowercase letters represented the significant difference among different distances (*p* < 0.05).

**Table 2 ijerph-20-03689-t002:** The topological property parameters of networks.

Network Indexes	AMF			Soil Fungi		
	100 m	900 m	1900 m	100 m	900 m	1900 m
Nodes	39	48	47	140	155	162
Edges	41	70	71	1000	1649	1717
Positive links (%)	59.52	66.2	52.78	48.75	50.12	50.23
Negative links (%)	40.48	33.8	47.2	51.25	49.88	49.77
avgK	2.154	2.958	3.064	14.3	21.29	21.21
avgCC	0.057	0.13	0.046	0.114	0.168	0.157
GD	4.45	3.59	3.326	2.121	1.938	1.951
Modularity	0.614	0.513	0.47	0.214	0.177	0.17

AMF, arbuscular mycorrhizal fungi; avgK, average degree; avgCC, average clustering coefficients; GD, average path distances.

## Data Availability

Not applicable.

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
