# Peer review of "Effect of Coal Mining on Soil Microorganisms from Stipa krylovii Rhizosphere in Typical Grassland"

_ijerph, 2023, doi:10.3390/ijerph20043689_

Round 1

Reviewer 1 Report

Dear authors,

Your paper is very interesting; however, I would like to point out some weak points. 

Line 153: bray-Bray (correct it)

Line 158: mantel-Mantel

In which database did you upload your sequences? This must be stated in method section. 

Table 1: which statistical test was used for significance testing? I did not notice it, neither in the method section neither in results. 

Results: please describe which statistical test was used for significance analyzing. I suggest that you describe in method section more detailed, which statistical test were used for significance analyzing, because it's an important information. 

Figure 2: usage of different colors in figure is a bit disturbing as I was looking for the legend. I suggest that you use one color as you have labels on x-axis and different colors for boxplots do not give any additional information's, in contrast they confuse the reader.  Line for significance/non-significance of the results are hard to read, I would suggest that you use different letter marks (e.g. Tukey contrasts), as the reader would recognize immediately the significance of the results.   

Figure 3: B-F again, usage of different colors is not needed. Use one color. And use different letters for showing the significance of the results.

Author Response

Reviewer #1

Comment 1: Line 153: bray-Bray (correct it).

Response: Done.

Comment 2: Line 158: mantel-Mantel.

Response: Done.

Comment 3: In which database did you upload your sequences? This must be stated in method section.

Response: DNA sequences were compiled and deposited in GenBank database and stated it in method section in the revised manuscript.

Comment 4: Table 1: which statistical test was used for significance testing? I did not notice it, neither in the method section neither in results.

Response: Thanks, agree and the statistical test used for significance testing have been added in “2.4 Data analyses”. Please see Lines 178-181 in the revised manuscript.

Comment 5: Results: please describe which statistical test was used for significance analyzing. I suggest that you describe in method section more detailed, which statistical test were used for significance analyzing, because it's an important information. Response: Thanks to the reviewer’s suggestion, the statistical test used for significance testing have been added in “2.4 Data analyses” of method section.

Comment 6: Figure 2: usage of different colors in figure is a bit disturbing as I was looking for the legend. I suggest that you use one color as you have labels on x-axis and different colors for boxplots do not give any additional information's, in contrast they confuse the reader. Line for significance/non-significance of the results are hard to read, I would suggest that you use different letter marks (e.g. Tukey contrasts), as the reader would recognize immediately the significance of the results.  

Response: According to the reviewer's suggestion, one color and different letter marks in boxplots have been changed in Figure 2.

Comment 7: Figure 3: B-F again, usage of different colors is not needed. Use one color. And use different letters for showing the significance of the results.

 Response: Agree, one color and different letter marks in boxplots have been changed in Figure 3. Thanks again.

Reviewer 2 Report

I can suggest minor revision in order to clarify some points of study.

Although the research topic and some of results interesting, no in depth discussions were given to show relationship with microbial communities. If there is no suspected correlation, the authors should state so. However, that would seem to be rather abrupt.  

The abstract should be specific and scientific information.

Author Response

Reviewer #2

Comment 1: Although the research topic and some of results interesting, no in depth discussions were given to show relationship with microbial communities. If there is no suspected correlation, the authors should state so. However, that would seem to be rather abrupt. 

Response: According to the reviewer's suggestion, the results, discussion and conclusions of the article have been major revised, we hope that the revised manuscript can meet the requirements of publication. Thanks for your suggestion.

Comment 2: The abstract should be specific and scientific information.

Response: Thanks for the suggestions of reviewer, we have revised the abstract in the manuscript, all the corrections and changes are marked in red. Thanks again.

Reviewer 3 Report

Ms. No.: ijerph-2043038

Title: Effect of coal mining on soil microorganisms from Stipa krylovii rhizosphere in typical grassland

There are several issues need to be revised. The study designed three sampling sites at different distances (100 m, 900 m, and 1900 m) from the mining area to collect samples and compare the differences in microbial composition and ecological network of the samples affected by mining. However, the three sampling sites themselves already have different soil characteristics parameters, how to distinguish whether the differences in their microbial composition and ecological networks are influenced by mining or by differences in their own soil characteristics parameters? Besides, only three samples are not enough to explain the problem. Also, which sample is the control of the article, the 1900 m sample? How can it be used as a control when its soil characteristics are different from other samples?

The amount of data in the article is too small, and the given information is not attractive enough. Data in “3.5 AMF and soil fungal networks” and “3.6 Correlation analysis” are not fully explored, especially networks sequencing can often offer a lot of data, but they are not shown in this article.

The Discussion part is not deep enough, and not tightly enough to the topic, many contents are less related to the experimental results, and the relevant literature cited is few. In particular, the data in “3.5 AMF and soil fungal networks” and “3.6 Correlation analysis” have not been fully explored. As a result, the Conclusions and the Abstract also show few important information.

There are several language problems, e.g., “Abstract: Coal mining affects plants and soil, ultimately it alters microecological health.”, “but...” in line 71.

The article should be in the passive rather than active voice, e.g. Abstract: we analyzed; We also determined; Our results..... Besides, “We would ...” in line 76; “We analyzed” in line 92.

Author Response

Reviewer #3

Comment 1: The study designed three sampling sites at different distances (100 m, 900 m, and 1900 m) from the mining area to collect samples and compare the differences in microbial composition and ecological network of the samples affected by mining. However, the three sampling sites themselves already have different soil characteristics parameters, how to distinguish whether the differences in their microbial composition and ecological networks are influenced by mining or by differences in their own soil characteristics parameters? Besides, only three samples are not enough to explain the problem. Also, which sample is the control of the article, the 1900 m sample? How can it be used as a control when its soil characteristics are different from other samples?

Response: According to the early investigation of this study, the soil characteristics and plants at 1900 m and natural grassland beyond 2 km had no significant difference. Therefore, 1900 m sample was selected as the control in this study.

Comment 2: The amount of data in the article is too small, and the given information is not attractive enough. Data in “3.5 AMF and soil fungal networks” and “3.6 Correlation analysis” are not fully explored, especially networks sequencing can often offer a lot of data, but they are not shown in this article.

Response: The reviewer's suggestions are undoubtedly correct. In the network analysis section of the article "3.5 AMF and soil fungal networks," deeper data mining work (such as network modularization) should indeed be conducted. However, in this article, we tend to use conventional network data obtained from analysis to reflect the impact of coal mining activities on the stability of microbial communities, so we use indicators such as "Positive links" and "Modularity" to characterize. In future studies, we will combine multi-year continuous monitoring data to increase the data volume and deeply mine network data to explore the relative relationships between microbes in detail. We have expanded Section 3.6 based on the suggestions of the reviewers, and have added the relationship between coal mining disturbance and its impact on various factors. Please see Lines 316-323 in the revised manuscript.

Comment 3: The Discussion part is not deep enough, and not tightly enough to the topic, many contents are less related to the experimental results, and the relevant literature cited is few. In particular, the data in “3.5 AMF and soil fungal networks” and “3.6 Correlation analysis” have not been fully explored. As a result, the Conclusions and the Abstract also show few important information.

Response: The suggestions of the reviewer are undoubtedly correct, and we have made significant revisions and improvements to the discussion section by delving into the results of sections 3.5 and 3.6, based on the reviewer's suggestions. Thank you again for the valuable suggestions of the reviewer.

Comment 4: There are several language problems, e.g., “Abstract: Coal mining affects plants and soil, ultimately it alters microecological health.”, “but...” in line 71.

Response: Considering the reviewer’s suggestion, language problems in “Abstract” and “Introduction” sections have been corrected. Please see Lines 12-13 and 79        in the revised manuscript.

Comment 5: The article should be in the passive rather than active voice, e.g. Abstract: we analyzed; We also determined; Our results..... Besides, “We would ...” in line 76; “We analyzed” in line 92.

Response: Thanks. Agreed and we have changed accordingly description in “Abstract” and “Introduction” as suggested. Please see Lines 18, 21 and 86-87 in the revised manuscript.

Reviewer 4 Report

[IJERPH] Manuscript ID: ijerph-2043038

Title: Effect of coal mining on soil microorganisms from Stipa 2 krylovii rhizosphere in typical grassland

In my opinion, this study is interesting and timely and has the potential to be published by IJERPH – MDPI. However, some doubts should be solved, and a few points must be clarified before this occurs. These considerations are described below:

General comment:

1)      The manuscript is well structured, but some constructions made in English could be improved to make its content more precise. There are also some repetitions of terms in the same sentences or paragraphs that could be adjusted to make the reading more fluid. Therefore, I suggest the document be revised regarding these aspects of expression.

Introduction:

2)      The Introduction is well-written concerning the context in which the research takes place and the gap it intends to fill. However, I suggest that authors avoid lumping ideas such as those found in references [11 – 15], [25 – 27], and [30 – 32]. That is: It would be more convenient if each citation were linked to only an idea.

3)      Lines 92-94: I suggest removing the following passage: 'We analyzed the network structures of AMF and fungal communities to determine if different fungal community responds differently to coal mining.' At this point in the text, you are presenting the research objectives, and this content refers to the method used to achieve them. If you wish to maintain the structure, I recommend you do so after specifying the objectives, including more details about how the research was conducted, even if this occurs synthetically.

4)      I suggest the Introduction section be completed with information about the gains this investigation can bring and which audience would benefit from these findings.

Material and Methods:

General comment: It is essential to explain the reasons that led you to carry out analyzes of soil pH, soil DNA, QIIME, and others, as a means of obtaining the results that made the objectives of the investigation to be achieved. That is: Why were these analyzes (and only these) selected? Because the conclusions that the results they can provide help to meet the objectives designed for the study?

5)      Lines 107-113: This passage is confusing, perhaps because of the repetition of ideas. Please review its content.

6)      Line 108: What criteria were used to define the sampling areas (routes A, B, and E)? Are they representative of conditions in the region under study? Did you apply any statistical method to define the number of samples to be collected?

Results:

7)      General comments: You have generated many results from the investigation and analyses you have carried out. This is very important and makes the study consistent. However, when reading and reflecting on the contents of sections 3.1 through 3.5, I noticed an overlap between the description of these findings and the accompanying graphs and tables. What I mean by this is that a (large) portion of the text describes the same results seen in the graphs and tables. This may have occurred because you separated the Results and Discussion sections. Unfortunately, this decision led to more problems. In addition to making reading these passages quite tricky, it was necessary to go back to them on many occasions when I had already reached the Discussion (section 4). Having to act that way, I had difficulty integrating both contents. I suggest that the discussion of each aspect of the analysis is associated with the presentation of results, thus solving this problem of fluidity. However, for this action to be efficient, the text needs to be readjusted to comment (only) on the crucial aspects of the research without becoming tiring.

8)      Figures 1a – 1c: I had difficulty understanding these diagrams without the help of the text described in section 3.2. One of the reasons for this to have happened was that the figures were small and, therefore, difficult to read. So, I suggest that they be revised in terms of size and, eventually, how those results are presented.

9)      Figures 2 and 3: The earlier comment about the size of the figures in the manuscript also applies to Figures 2 and 3. In these cases, however, although small, the diagrams use a conventional and quite clear form of expression. In any case, their sizes must be adjusted.

Discussion:

10)   I expected more extensive and consistent discussions, given that you generated many results (see comment n. 7 of this review). However, I understand that there are also clear points of relationship between the different strands of analysis that could be better explored. In addition, the Introduction section showed that there are works in the same field in which this investigation took place. Why didn't you try to make correlations with the literature, even if these were only for certain parts of the research? This framework leads to a research framing problem that needs to be adjusted.

Conclusions:

11)   It would be important that in addition to the findings, aspects such as limitations of the investigation, future developments of this study, and applications to which its results are intended were part of the Conclusions section (I had already commented on this same aspect in topic 4 of this review).

Author Response

Reviewer #4

Comment 1: The manuscript is well structured, but some constructions made in English could be improved to make its content more precise. There are also some repetitions of terms in the same sentences or paragraphs that could be adjusted to make the reading more fluid.

Response: According to the reviewer's suggestion, the results, discussions and conclusions of the article have been major revised, we hope that the revised manuscript can meet the requirements of publication. Thanks for your suggestion.

Comment 2: The Introduction is well-written concerning the context in which the research takes place and the gap it intends to fill. However, I suggest that authors avoid lumping ideas such as those found in references [11-15], [25-27], and [30-32]. That is: It would be more convenient if each citation were linked to only an idea.

Response: According to the reviewer's suggestion, references [11-15], [25-27], and [30-32] have been revised, each citation have been linked to one or two ideas. Please see Lines 64, 82-83 and 85-86 in the revised manuscript.

Comment 3: Lines 92-94: I suggest removing the following passage: 'We analyzed the network structures of AMF and fungal communities to determine if different fungal community responds differently to coal mining.' At this point in the text, you are presenting the research objectives, and this content refers to the method used to achieve them. If you wish to maintain the structure, I recommend you do so after specifying the objectives, including more details about how the research was conducted, even if this occurs synthetically.

Response: Thanks. Agreed and we have removed this passage as suggested.

Comment 4: I suggest the Introduction section be completed with information about the gains this investigation can bring and which audience would benefit from these findings.

Material and Methods:

General comment: It is essential to explain the reasons that led you to carry out analyzes of soil pH, soil DNA, QIIME, and others, as a means of obtaining the results that made the objectives of the investigation to be achieved. That is: Why were these analyzes (and only these) selected? Because the conclusions that the results they can provide help to meet the objectives designed for the study?

Response: Thanks for the suggestions of reviewer, we have added the reasons for selecting these measurement indicators in “Materials and Methods” section. Please see Lines 133-181 in the revised manuscript. Thanks again.
Comment 5: Lines 107-113: This passage is confusing, perhaps because of the repetition of ideas. Please review its content.

Response: Yes, repetition ideas have been removed, thank you for your suggestion.

Comment 6: Line 108: What criteria were used to define the sampling areas (routes A, B, and E)? Are they representative of conditions in the region under study? Did you apply any statistical method to define the number of samples to be collected?

Response: Thank you for the reviewer's questions. The main basis for selecting the sampling area is the radiation extension in different directions from the extended mining area. By taking samples along three different directions of radiation, we aim to minimize the impact of other environmental factors on the experimental results. At the same time, we try to meet the statistical requirements (more than three samples) for the number of samples in one direction of sampling. Therefore, we collected a total of 12 samples (3 lines x 4 repetitions) on a distance gradient.

Comment 7: Results: General comments: You have generated many results from the investigation and analyses you have carried out. This is very important and makes the study consistent. However, when reading and reflecting on the contents of sections 3.1 through 3.5, I noticed an overlap between the description of these findings and the accompanying graphs and tables. What I mean by this is that a (large) portion of the text describes the same results seen in the graphs and tables. This may have occurred because you separated the Results and Discussion sections. Unfortunately, this decision led to more problems. In addition to making reading these passages quite tricky, it was necessary to go back to them on many occasions when I had already reached the Discussion (section 4). Having to act that way, I had difficulty integrating both contents. I suggest that the discussion of each aspect of the analysis is associated with the presentation of results, thus solving this problem of fluidity. However, for this action to be efficient, the text needs to be readjusted to comment (only) on the crucial aspects of the research without becoming tiring.

Response: Thanks for the suggestions of reviewer, some summary statements after each “Result” and the corresponding chart information in the “Discussion” section have been added to increase the readability of the article.

Comment 8: Figures 1a-1c: I had difficulty understanding these diagrams without the help of the text described in section 3.2. One of the reasons for this to have happened was that the figures were small and, therefore, difficult to read. So, I suggest that they be revised in terms of size and, eventually, how those results are presented.

Response: According to the reviewer's suggestion, Figures 1a -1c have been changed   in the revised manuscript.

Comment 9: Figures 2 and 3: The earlier comment about the size of the figures in the manuscript also applies to Figures 2 and 3. In these cases, however, although small, the diagrams use a conventional and quite clear form of expression. In any case, their sizes must be adjusted.

Response: Thanks for your suggestions, the size, color, and different letter marks in boxplots have been changed in Figures 2 and 3.

Comment 10: Discussion: I expected more extensive and consistent discussions, given that you generated many results (see comment n. 7 of this review). However, I understand that there are also clear points of relationship between the different strands of analysis that could be better explored. In addition, the Introduction section showed that there are works in the same field in which this investigation took place. Why didn't you try to make correlations with the literature, even if these were only for certain parts of the research? This framework leads to a research framing problem that needs to be adjusted.

Response: According to the reviewer's suggestion, the research results of previous researchers are supplemented, and linked references and expanded with our research results in the “Discussion”, thanks again.

Comment 11: It would be important that in addition to the findings, aspects such as limitations of the investigation, future developments of this study, and applications to which its results are intended were part of the Conclusions section (I had already commented on this same aspect in topic 4 of this review).

Response: Thanks for the suggestions of reviewer, directions of future research have been supplemented in the “Conclusions”. Please see Lines 470-476 in the revised manuscript.

Round 2

Reviewer 1 Report

Dear authors,

I checked the revised manuscript and believe that is now much better. 

Reviewer 3 Report

The paper can be accepted for publication.